# Crosstalk Between Adipokines and Paraoxonase 1: A New Potential Axis Linking Oxidative Stress and Inflammation

**DOI:** 10.3390/antiox8080287

**Published:** 2019-08-06

**Authors:** Veronica Tisato, Arianna Romani, Elisa Tavanti, Elisabetta Melloni, Daniela Milani, Gloria Bonaccorsi, Juana M. Sanz, Donato Gemmati, Angelina Passaro, Carlo Cervellati

**Affiliations:** 1Department of Morphology, Surgery and Experimental Medicine and LTTA Centre, University of Ferrara, 44121 Ferrara, Italy; 2Department of Biomedical and Specialist Surgical Sciences, Section of Medical Biochemistry, Molecular Biology and Genetics, University of Ferrara, 44121 Ferrara, Italy; 3Department of Biomedical and Specialist Surgical Sciences and LTTA Centre, University of Ferrara, 44121 Ferrara, Italy; 4Department of Morphology, Surgery and Experimental Medicine, Menopause and Osteoporosis Centre, University of Ferrara, 44121 Ferrara, Italy; 5Department of Medical Sciences, Internal Medicine and CardioRespiratory Section, University of Ferrara, 44121 Ferrara, Italy; 6Department of Biomedical and Specialist Surgical Sciences, Section of Medical Biochemistry, Molecular Biology and Genetics-Ctr. Hemostasis & Thrombosis, University of Ferrara, 44121 Ferrara, Italy; 7University Center for Studies on Gender Medicine, University of Ferrara, 44121 Ferrara, Italy

**Keywords:** oxinflammation, paraoxonase 1 (PON1), adipokines, resistin, postmenopausal woman

## Abstract

Paraoxonase 1 (PON1) is a high-density lipoprotein (HDL)-associated protein that endows its carrier with (lipo-)lactonase-dependent antioxidative features. Low levels of PON1 activity have been observed in association with obesity, a major risk factor for cardiovascular disease (CVD). Considering the well-recognized atheroprotective role of PON1, exogenous/endogenous factors that might modulate its levels/activity are raising great interest. Since adipokines represent a molecular link between obesity and CVD, we here explored the possible impact of these substances on PON1 activity/expression. The levels of interleukin (IL)-6, IL-8, tumor necrosis factor alpha, monocyte chemoattractant protein-1, hepatocyte growth factor, resistin, leptin, and adiponectin were measured along with arylesterase, paraoxonase, and lactonase activities of PON1 in 107 postmenopausal women. Moreover, the direct effect of resistin on *PON1* expression was evaluated in vitro. Multivariate analysis revealed that only resistin was significantly and inversely correlated with PON1-lactonase activities (*r* = −0.346, *p* < 0.001) regardless of confounding factors such as age or HDL-cholesterol. It is worth noting that no statistical link was found between adipokine and arylesterase or paraoxonase, the two promiscuous activities of PON1. Notably, resistin down-regulated *PON1* expression occurred in hepatocellular carcinoma cultures. Our study suggests that resistin might be a negative modulator of PON1 expression and anti-oxidative activity.

## 1. Introduction

Paraoxonase 1 (PON1) (E.C. 3.1.8.1) is a member of a family of enzymes secreted by the liver and detectable as circulating molecules in the bloodstream bound to high-density lipoproteins (HDL) and, to a minor extent, to other lipoproteins [1,2]. PON1 possesses one of the broadest substrate specificities known; it exerts three hydrolytic activities [1]: (1) paraoxonase activity towards toxic organophosphates such as paraoxon (diethyl p-nitrophenyl phosphate), the toxic oxon metabolite of parathion, an insecticide; (2) arylesterase activity towards non-phosphorous arylesters; (3) lactonase activity towards several lactones, including lipo-lactones generated after fatty acid oxidation.

A wealth of evidence highlights the role of PON1 in the anti-inflammatory and antioxidant properties ascribed to HDL [3]. In particular, a high content in PON1 appeared to enhance the ability of these lipoproteins to prevent peroxidative damage to low-density lipoproteins (LDL) and cell membranes (which resemble LDL), and it has been suggested that they may protect against pro-inflammatory activation of macrophages and endothelium [4,5,6]. On the contrary, low PON1 activity seems to result in a decreased functional efficiency of HDL, which in turn increases the risk of developing inflammatory diseases, including (but not limited to) atherosclerosis [5,7,8].

Accumulation of fat, especially at the intra-abdominal level, is a well-established risk factor for cardiovascular disease (CVD) and several metabolic diseases [9]. The mechanisms linking obesity to clinical presentations involve a self-perpetuating loop between chronic low-grade inflammation and oxidative stress (OxS) [10]. The effect of this axis becomes health-threatening when defensive/compensating mechanisms are affected/impaired. This event occurs for example in the context of obesity, a condition in which antioxidant and anti-inflammatory agents are not able to effectively counteract the persistence of a source of toxicants, reactive species, and pro-inflammatory cytokines [11]. PON1 is a paradigmatic example of the failure of compensatory mechanisms, and decreased enzyme activity has been indeed found in obese subjects in the presence of CVD and in obese subjects free of clinical manifestations of CVD [12,13,14,15]. The unproductive protective effect of this HDL-accessory protein may result in exacerbated vessel damage and in general to the establishment of pro-atherosclerotic processes [1,7]. The overall effects of this sequence of events are confirmed by the epidemiological evidence of an inverse correlation between PON1 activity and incidence of coronary events in individuals with either low or high risk of CVD [16,17].

In this scenario, the biological mechanisms underlying the decreased activity of PON1 in obesity and obesity-related pathological conditions still need to be fully clarified. A proposed link between low PON1 activity and an increased risk of diabetes/metabolic disease refers to the potential effects of downstream and/or upstream mediators of three key pathways, such as OxS, metabolism, and inflammation [2,18].

Adipokines are bioactive molecules released by (white) adipose tissue that couple the regulation of energy metabolism, insulin sensitivity, inflammation, and atherogenesis, thereby linking obesity with CVD [19,20]. Excess of adiposity is associated with a dramatic perturbation in adipokines secretion, with increase of pro-inflammatory cytokines (e.g., interleukin 6 (IL-6), IL-8, tumor necrosis factor *α* (TNF-*α*), leptin, and resistin) and a concomitant decrease in anti-inflammatory molecules, such as adiponectin [21]. In women, this detrimental change in circulating adipokines pattern typically occurs after menopause, due to a gynoid to android shift in body fat distribution [22,23]. It is well-known that the preferential visceral/abdominal fat accumulation instead of gluteo-femoral accumulation, is associated with a significant increase in risk of metabolic disturbance and CVD [24]. Since adipokines play key roles in mediating the detrimental effects of fat excess particularly on vascular integrity, it is tempting to hypothesize that in obese individuals there might be a link between adipokines alteration and the observed PON1 deficit. Of note, although this axis is of great interest in a clinical perspective, insights on this potential interconnection still need to be fully elucidated.

On these bases, the aim of the present study is to investigate the potential crosstalk between altered adipokines levels and PON1 activity in a cohort of apparently healthy post-menopausal women, hypothesizing potential mechanistic links that could account for the clinical and epidemiological observations.

## 2. Materials and Methods 

### 2.1. Study Participants

Subjects included in the study (*n* = 107) were randomly selected from women attending the Menopause and Osteoporosis Center of the University of Ferrara (Italy). This cohort is part of a larger sample (*n* = 513) enrolled in the frame of a research protocol aiming to explore the effect of body fat distribution and obesity on metabolic, oxidative, and inflammatory profiles in women with pre-, peri-, and postmenopausal status [25]. The study was carried out in accordance with the Declaration of Helsinki (World Medical Association, http://www.wma.net) and approved by the ethics committee of the University-Hospital of Ferrara. Each subject signed an informed consent form to enter the study. Eligible participants were Caucasian, apparently healthy postmenopausal women between 50 and 65 years old. Menopausal status was defined according to ReSTAGE’s modification of Stages of Reproductive Aging Workshop (STRAW) as described elsewhere [23]. Women were not included in the study in case of self-reported excessive alcohol intake (more than 20 g/day), pharmacological/hormonal treatments, concomitant chronic kidney or liver diseases, CVD, and cancer.

### 2.2. Biochemical Assays

Peripheral blood samples were collected by venipuncture into Vacutainer tubes without anticoagulant after overnight fast. After 30 min of incubation at room temperature, blood samples were centrifuged at 4.650 g for 20 min, and sera were stored at −80 °C until analysis. Samples were frozen and thawed only once before performing the biochemical assays.

### 2.3. Adipokines Assessment

Adipokines were assessed in sera as previously described [26] by using the MILLIPLEX MAP Human Panels (Merck Millipore, Billerica, MA). These bead-based multiplex immunoassays allow simultaneous quantification of the following human adipokines: adiponectin (CV: intra-assay = 4%, inter-assay = 10%), resistin (CV: intra-assay = 3%, inter-assay CV = 14%), IL-6 (CV: intra-assay = 2%, inter-assay = 10%), IL-8 (CV: intra-assay = 3%, inter-assay CV = 14%), leptin (CV: intra-assay CV = 5%, inter-assay = 13%), monocyte chemoattractant protein 1 (MCP-1) (CV: intra-assay = 2%; inter-assay = 11%), tumor necrosis factor *α* (TNF-α) (CV: intra-assay = 3%; inter-assay = 16%), and hepatocyte growth factor (HGF) (CV: intra-assay = 3%; inter-assay = 11%). Samples were processed in duplicate following the manufacturer’s recommended protocols and read on a MAGPIX instrument (Merck Millipore) equipped with the MILLIPLEX-Analyst Software (Merck Millipore) using a five-parameter nonlinear regression formula to compute sample concentrations from the standard curves. Quality controls provided in the multiplex kits were used to validate the assay performance.

### 2.4. PON1 Activities

Serum lactonase, paraoxonase, and arylesterase activities of PON1 were measured by UV–VIS spectrophotometric assays in a 96-well plate by using a Tecan Infinite M200 microplate reader (Tecan Group Ltd., Männedorf, Switzerland).

Lactonase activity was assessed by using gamma-thiobutyrolactone (TBL, Sigma-Aldrich) as a substrate, and Ellman’s procedure was used to spectrophotometrically monitor (412 nm) the accumulation of free sulfhydryl groups via coupling with 5,5-dithiobis(2-nitrobenzoic acid) (DTNB), as described elsewhere [27]. The reaction was started after introducing serum to the reaction mixture containing buffer (50 mmol/L Tris, 1 mM CaCl_2_, 50 mmol NaCl, pH = 8), 0.5 mmol/L DTNB (Sigma-Aldrich), and 10.5 mmol/L TBL in each well. A molar extinction coefficient of 13,600 M^−1^ cm^−1^ was used for the enzymatic unit determination (expressed as μmol/L/min). 

PON1 activity towards paraoxon was evaluated by assessing (at 405 nm) the rate of formation of para-nitrophenol, in a reaction mixture composed of 1.5 mM paraoxon (Sigma-Aldrich), 10 mM Tris-HCl, pH = 8, 0.9 M NaCl, and 2 mM CaCl_2_. An extinction coefficient of 17,000 M^−1^ cm^−1^ was used for calculating units of paraoxonase activity (expressed as μmol/L/min).

Arylesterase activity was measured at 270 nm after adding 10 µL of sample to 240 µL of reaction mixture composed of 1 mM phenylacetate, 9 mM Tris-HCl, pH = 8, and 0.9 mM CaCl_2_ (temperature: 24 °C). An extinction coefficient of 1310 M^−1^ cm^−1^ was used for calculating units (KU) of arylesterase activity (expressed as μmol/L/min).

### 2.5. Cell Treatments and RT-PCR Analysis

Human hepatocellular carcinoma HEP 3B2.1-7 cell line was purchased from American Type Culture Collection (ATCC, Manassas, VA, USA). Cells were cultured in Dulbecco’s modified Eagle’s medium, supplemented with 10% fetal bovine serum, L-glutamine, and antibiotics (all from Gibco, Grand Island, NY) and maintained at 37 °C and 5% CO_2_. Cells were seeded at a density of 100.000/well in a 6-well plate 24 h prior to drug treatments. Cells were treated with recombinant IL-6 at the optimal concentration of 10 ng/ml, determined in preliminary dose-response experiments, used alone or in combination with 25 or 100 ng/ml of recombinant resistin (both from Peprotech, Rocky Hill, NJ, USA) in serum-free medium. In the combined treatments, IL-6 was added to the cells at the indicated concentration 1 h before treatment with resistin. Cell viability was monitored by trypan blue dye exclusion. Total RNA was extracted by using the miRNeasy Mini kit (QIAGEN, Hilden, Germany) by following the manufacturer’s instructions and as previously described [28]. Briefly, after genomic DNA removal with the RNase-Free DNase Set (QIAGEN), 200 ng of RNA were retrotranscribed and amplified using the Express One-Step Superscript qRT-PCR Kit, universal (Thermo Fisher Scientific, Rockford, IL) with TaqMan assay technology. The POLR2A was used as the housekeeping gene. The TaqMan assays used were Hs00166557_m1 (for PON1) and Hs00172187_m1 (for POLR2A).

### 2.6. Statistical Analysis

Correlations among epidemiological data were examined by Pearson’s and Spearman’s tests (for normally and non-normally distributed variables, respectively). Multiple regression analysis was used to determine if associations were independent of potential confounding factors. Statistical significance was defined as *p* < 0.05.

The ethics committee of the University of Ferrara approved the two research studies (REC numbers: CE 110291 and CE 130292) from which the data presented here were collected. The research protocol was carried out in accordance with the Declaration of Helsinki (World Medical Association, http://www.wma.net). Written informed consent was obtained from each patient during the first office visit at baseline before the possible inclusion in the study.

## 3. Results

### Associations Between PON1 Activities and Adipokines: Study Population and In Vitro Assessment

The main clinical characteristics of the cohort of women enrolled in the study and the levels of the measured adipokines and PON1 activities are shown in Table 1 and Appendix A, respectively.

Potential associations between adipokine levels and PON1 activities were first examined by univariate analysis. As displayed, arylesterase showed only a positive association with IL-8 (*p* = 0.006), while no significant correlations were detected for PON1-paraoxonase activity. Lactonase activity was found to be correlated with resistin (*p* = 0.0007) (Table 2 and Appendix A).

To evaluate whether the emerged univariate correlations were independent of potential confounding factors, we performed a multiple regression analysis including the following covariates: age, HDL-C, hypertension, smoking, waist circumference, and Hs-CRP (Table 3). Body mass index (BMI) was excluded from the covariate set because of its collinearity with waist circumference (of note, the inclusion of BMI in place of waist circumference did not affect multiple regression test outcomes). As shown in Table 3, we found that arylesterase activity was no longer associated with IL-8 after full adjustment. On the contrary, the association involving lactonase retained its significance with no evident changes in the regression coefficient strength (lactonase vs. resistin, *p* < 0.001) (Table 3).

Finally, to assess whether resistin might be directly involved in the downregulation of PON1, we carried out in vitro experiments by using human hepatocellular carcinoma cells as a hepatocytes model, which are known to express PON1 and are thought to significantly contribute to the release of PON1 in the systemic circulation [29]. Since it has been previously demonstrated that IL-6 increases *PON1* expression at the transcriptional level [30,31], we assessed the hypothesized effect of resistin on *PON1* expression on HEP 3B2.1-7 cells. As shown in Figure 1, cell treatment with recombinant resistin resulted in the down-modulation of IL-6 induced PON1 mRNA levels in a dose-dependent manner. On the contrary, in the absence of an inflammatory milieu, resistin-mediated down-modulation was detectable only at the highest dose of resistin used (Figure 1). The evidence that resistin and IL-6 had negligible effects on cell viability ruled out the possibility that PON1 down-modulation was a mere indirect consequence of the cytotoxic effect of resistin and/or IL-6.

## 4. Discussion

The term “Oxinflammation” is a recently proposed concept describing the vicious cycle involving systemic OxS and low-grade chronic inflammation representing a subclinical feature of a number of pathologies, such as diabetes and CVD [11,32]. The current knowledge on PON1 suggests that this enzyme may be a potential “chain-breaker” of this deleterious self-perpetuating process. Indeed, PON1 was reported to prevent the formation of oxidized LDLs (OxLDLs), a well-recognized link between oxidative stress and immune responses [5,33,34,35]. Modified lipoproteins are then able to enhance reactive species production and trigger the expression of pro-inflammatory cytokines and chemokines in macrophages, vascular smooth muscle cells, and endothelial cells [2,4]. Given the proposed important role of PON1 in atherosclerosis, it is reasonable to hypothesize that its hepatic production and release in the blood circulation should be finely regulated by endogenous factors, in particular by those playing a role in disease pathogenesis. Adipokines are bioactive molecules released by adipose tissue and known to be differentially involved in obesity, diabetes, CVD, and other pathological conditions [21]. In this light, we investigated potential associations between a panel of these bioactive substances and PON1 in postmenopausal healthy women. Within the adipokines analyzed, only resistin emerged from the multivariate analysis as an independent and negative predictor of lactonase but not of paraoxonase or arylesterase PON1 activity. To explore the possible mechanism underlying this epidemiological evidence, we evaluated for the first time the ability of resistin to modulate PON1 expression on hepatic HEP 3B2.1-7 cells, and we found that this adipokine is able to downregulate PON1 at a transcriptional level in this model. 

The panel of adipokines considered in this study is critically involved in the regulation of redox processes, immune responses, and inflammation. At first glance, the absence of significant relationships between these substances and arylesterase PON1 activity (i.e., the promiscuous PON1 activity which better reflects PON1 concentration because of its low inter-individual variability [36,37]), seems in conflict with the current literature. Nevertheless, from a more in-depth analysis, despite compelling data coming from in vitro models, results obtained in human/clinical studies are overall still scarce and highly inconsistent.

With regard to a potential axis between adipokines and PON1, intriguing data come from preclinical experimental studies on IL-6 and TNF-α, showing that these two cytokines upregulate and downregulate *PON1* gene expression respectively, suggesting that alterations in cytokines/adipokines levels might affect PON1 activity [30,31]. However, although providing valuable mechanistic insights, in vitro models cannot perfectly resume the in vivo settings, also for the supra-physiological concentrations of the cytokines and biological factors employed in the experiments. Moreover, the impact of PON1 could be “tangible” only in selected (physio) pathological contexts realistically negligible at a systemic level. In this scenario, compelling studies showing the ability of PON1-containing HDL to modulate MCP-1 secretion from endothelial cells [33], as well as to suppress the pro-inflammatory responses of macrophages [38], may not result in a significant correlation between circulatory levels of the chemokines and PON1. Intriguing results were also obtained with leptin, which was shown to suppress PON1 activity via pro-oxidative and pro-inflammatory pathways [39,40]. However, epidemiological data highlight that this interaction may not reverberate at the systemic level and did not confirm the cross-talk [41,42]. In a similar fashion, studies on potential links between adiponectin and PON1 activity generated highly variable results [39,43,44]. Focusing on resistin, Bajnok and colleagues [45] failed to find a significant association between arylesterase and the adipokine in a sample of mostly obese men/women, whereas Bednarska-Makaruk and colleagues reported a negative correlation between the adipokine and PON1 arylesterase activity in older patients with dementia [46].

Resistin is expressed primarily in cells of monocyte/macrophage lineage and it is elevated in obese and diabetic individuals [47]. Resistin promotes insulin resistance, local and systemic inflammation and induces endothelial dysfunction and increased ROS synthesis in various cells with an overall pathogenic role in CVD and diabetes [48]. In our study, the observed lack and presence of an inverse correlation of the adipokine with arylesterase and lactonase, respectively, would suggest that an increase in resistin might not affect PON1 secretion from liver, but only its physiological activity. Indeed, if arylesterase activity is a well-recognized surrogate of PON1 concentration, (lipo)lactonase is now ascribed as the native/biological activity which may be responsible for the degradation of truncated oxidized fatty acids from phospholipid, cholesteryl ester, and triglyceride hydroperoxides [49]. In this scenario, it is difficult to hypothesize potential mechanisms underlying the downregulation of *PON1* expression induced by resistin observed in vitro, mainly because of the poor knowledge of PON1 biology and regulatory pathways. Nonetheless, an attempt in this direction can be made by following the conceptual trail suggested by Marsillach and colleagues [50]. By using a preclinical in vivo model based on experimental rats with chronic liver impairment induced by CCl4 administration leading to hepatic OxS and inflammation, Marsillach and colleagues found a decrease in *PON1* gene expression, which was in line with the decrease in serum lactonase, but not of esterase or PON1 serum levels (decreased only in the first week of treatment). Moreover, the reported increase in ROS concentration and inflammation in rat hepatocytes was suggested to be downstream of these events, being responsible for PON1 catabolism and secretion and for the inactivation of lactonase activity (described as more vulnerable to oxidative insults, compared to the other two promiscuous activities [51]). Thus, it is tempting to speculate that the observed crosstalk between resistin and PON1 could occur as an effect of oxinflammation onset in hepatic cells. 

## 5. Conclusions

PON1 is widely suggested to play a pivotal atheroprotective role, and therefore, a better understanding of the mechanisms by which PON1 can be modulated by internal/external factors has a strong potential clinical relevance. Our findings suggest that the pro-atherogenic and pro-inflammatory resistin might be a negative modulator of the biological activity of PON1. Although we are aware that the present study has some limitations mainly related to the degree of obesity in our cohort (that appears to be mild/moderate) and to the relatively low size of the sample, we here suggest that reduction in PON1 could represent one of the pathways linking obesity, adipokine secretion, and CVD. In this line, we believe that further studies are warranted to confirm our results in a different cohort of patients, also in a gender-perspective, and to elucidate the molecular mechanisms by which resistin may modulate PON1 activity and expression, providing a characterization that will bring new insight into a therapeutic clinical potential.

## Figures and Tables

**Figure 1 antioxidants-08-00287-f001:**
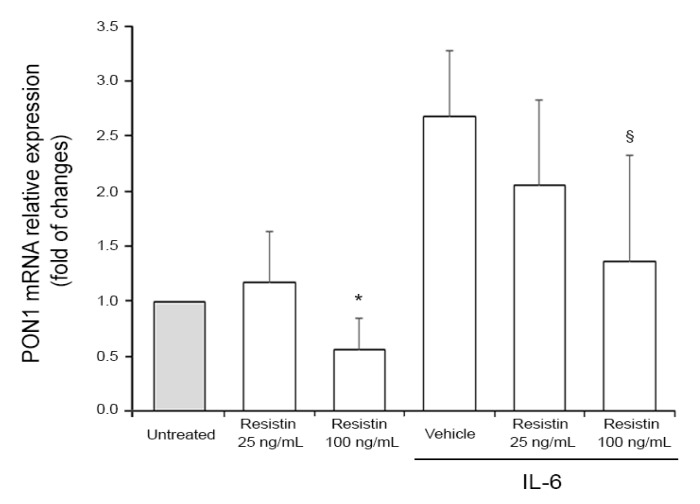
Resistin modulates *PON1* expression in vitro in HEP 3B2.1-7 cells. Cell cultures were exposed to human resistin (at the indicated concentrations) for 24 h alone or in combination with IL-6 (10 ng/mL). PON1 mRNA expression was analyzed by qRT-PCR and expressed as fold of change with respect to Untreated cultures set at 1. Results are reported as mean ± SD of three independent experiments. * *p* < 0.05 vs. untreated; ^§^*p* < 0.06 vs. vehicle.

**Table 1 antioxidants-08-00287-t001:** Principal characteristics of the subjects.

Demographic and Clinical Parameters	Value
Number of subjects, n	107
Age, years	57 ± 4
Years since menopause, years	4 (2–6)
Smoking, n (%)	8 (7)
Hypertension, n (%)	10 (9)
BMI, kg/m^2^	24 ± 3
Waist circumference, cm	82 ± 10
Total cholesterol, mmol/L	5.1 ± 1.3
HDL-C, mmol/L	1.8 ± 0.6
Triglycerides, mmol/L	0.9 ± 0.6
LDL-C, mmol/L	3.0 ± 1.3
hs-CRP, mg/dL	1.3 (0.5–3.0)

Data are expressed as % for categorical variables; mean ± standard deviation for normally distributed continuous variables; median (interquartile range) for non-normally distributed continuous variables. Abbreviations: BMI, body mass index; HDL-C, high density lipoprotein-cholesterol; LDL-C, low density lipoprotein-cholesterol; hs-CRP, high sensitivity-C reactive protein.

**Table 2 antioxidants-08-00287-t002:** Correlation coefficient between PON1 activities and adipokines.

Adipokines	Arylesterase	Paraoxonase	Log10 Lactonase
**IL-8**	0.265 ^a^	−0.014	0.086
**TNF-α**	0.001	0.090	−0.024
**IL-6**	−0.120	0.034	−0.039
**Resistin**	0.069	−0.102	−0.322 ^b^
**Adiponectin**	0.085	−0.049	−0.152
**Leptin**	0.017	0.164	0.070
**MCP-1**	0.022	0.012	−0.022
**HGF**	−0.037	0.053	−0.007

Abbreviations: IL (interleukin); TNF-α, tumor necrosis factor alpha; MCP-1, monocyte chemoattractant protein 1; HGF, hepatocyte growth factor a: *p* < 0.05; b: *p* < 0.001.

**Table 3 antioxidants-08-00287-t003:** Multiple regression analysis of the relationship between PON1-arylesterase and IL-8 and between PON1-lactonase activity and resistin.

Dependent Variable	Predictor	β (*p* Value)	β_age/HDL-C adjusted_ (*p* Value)	β_fully adjusted_ * (*p* Value)
Arylesterase	IL-8	0.265 ^a^	0.221	0.185
Lactonase	Resistin	−0.326 ^b^	−0.344 ^b^	−0.346 ^b^

* Covariates: age, HDL-C, waist circumference, smoking, hypertension, hs-CRP. Abbreviations: IL (interleukin). a: *p* < 0.05; b: *p* < 0.001.

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
