# Peer review of "Crosstalk Between Adipokines and Paraoxonase 1: A New Potential Axis Linking Oxidative Stress and Inflammation"

_antioxidants, 2019, doi:10.3390/antiox8080287_

Round 1

Reviewer 1 Report

The manuscript by Tisato and colleagues addresses the effects of cytokines/chemokines on paraoxonase 1 (PON1) levels and activity in postmenopausal women. Obesity is a risk factor for cardiovascular disease and may be associated with the loss of PON1 hydrolytic activities. Specifically, menopause is associated with an increase in visceral fat depots and alterations in levels of adipokines. The authors tested whether adipokines influence PON1 function. Adipokine levels were measured in serum of postmenopausal subjects using a multiplex assay. PON1 arylesterase, lactonase and paraoxonase activities were also measured. The results show that only resistin was inversely correlated with PON1 lactonase activity. There was no relationship between resistin and PON1 arylesterase or paraoxonase activities. Resistin also inhibited IL-6 induced PON1 expression in hepatocellular carcinoma cultures. The authors conclude that resistin is a negative modulator of PON1 expression and antioxidant activity.

Author Response

 We thank the reviewer for his/her comments. Following the reviewer's suggestion spelling and grammatical errors pointed  have been corrected

Reviewer 2 Report

Tisato and co-workers report results from a study investigating the possible effects of pro- and anti-inflammatory adipokines on the antioxidant PON1 enzyme activity associated with high-density lipoprotein.  Overall, the background and rationale for the study is clearly presented, the methods are appropriate, and the results/discussion are appropriate.  The authors' finding of a negative association of (pro-inflammatory) resistin specifically with PON1 lipolactonase activity, but with neither paraoxonase nor arylesterase activities, is novel and may potentially provide a link between previously demonstrated low PON1 activity seen in obese individuals.  The only suggested revision recommended for the authors is to consider including in their discussion some consideration of the possible limitations of their study.  In particular, the degree of obesity in the human cohort studied here seems relatively mild/moderate with BMI in the range (mean +/- 2SD) 18-30 kg/m2 and waist circumference in the range 62-102 cm (corresponding to  25-41 inches).

Author Response

We thank the reviewer for his/her comments. To address the reviewer's concerns regarding the limitations of the study we have re-written the conclusions and added some considerations about the actual caveats of our investigation (low size and degree of obesity) (see the last lines in the conclusions, page 10)

Besides, following the reviewer's suggestions,  spelling and grammatical errors have been corrected

Reviewer 3 Report

The Authors aimed to evaluate the potential crosstalk between altered adipokines levels and PON1 activity in a cohort of apparently healthy post-menopausal woman hypothesizing potential mechanistic links that could account the clinical and epidemiological observations. The Authors have investigated an interesting topic and the theme has been properly described. I would like to congratulate authors for the good-quality of the article, the literature reported used to write the paper, and for the clear and elegant and appropriate structure. The manuscript is well written, presented and discussed, and understandable to a specialist readership.

In general, the organization and the structure of the article are satisfactory and in agreement with the journal instructions for authors. The subject is adequate with the journal scope. The work shows a conscientious study in which a very exhaustive discussion of the literature available has been carried out. The introduction provides sufficient background, and the other sections include results clearly presented and analyzed exhaustively.

So, I recommend the acceptance of the paper.

Author Response

We thank the reviewer for his/her positive comments. Following the reviewer's suggestion spelling and grammatical errors pointed have been corrected